# In Silico Analysis of the Antagonist Effect of Enoxaparin on the ApoE4–Amyloid-Beta (A*β*) Complex at Different pH Conditions

**DOI:** 10.3390/biom12040499

**Published:** 2022-03-25

**Authors:** Jorge Alberto Aguilar-Pineda, Silvana G. Paco-Coralla, Camilo Febres-Molina, Pamela L. Gamero-Begazo, Pallavi Shrivastava, Karin J. Vera-López, Gonzalo Davila-Del-Carpio, Patricia López-C, Badhin Gómez, Christian L. Lino Cardenas

**Affiliations:** 1Laboratory of Genomics and Neurovascular Diseases, Vicerrectorado de Investigación, Universidad Católica de Santa María, Urb. San José s/n—Umacollo, Arequipa 04000, Peru; 70582393@ucsm.edu.pe (S.G.P.-C.); pshrivastava@ucsm.edu.pe (P.S.); kvera@ucsm.edu.pe (K.J.V.-L.); gdavilad@ucsm.edu.pe (G.D.-D.-C.); 2Centro de Investigación en Ingeniería Molecular—CIIM, Universidad Católica de Santa María, Urb. San José s/n—Umacollo, Arequipa 04000, Peru; pgamero@ucsm.edu.pe (P.L.G.-B.); bgomez@ucsm.edu.pe (B.G.); 3Doctorado en Fisicoquímica Molecular, Facultad de Ciencias Exactas, Universidad Andres Bello, Santiago 8370134, Chile; c.febresmolina@uandresbello.edu; 4Facultad de Ciencias Farmacéuticas, Bioquímicas y Biotecnológicas, Universidad Católica de Santa María, Urb. San José s/n—Umacollo, Arequipa 04000, Peru; 5Vicerrectorado de Investigación, Universidad Católica de Santa María, Urb. San José s/n—Umacollo, Arequipa 04000, Peru; plopez@ucsm.edu.pe; 6Cardiovascular Research Center, Cardiology Division, Massachusetts General Hospital, Boston, MA 02114, USA

**Keywords:** alzheimer disease, apolipoprotein E, amyloid-*β*, enoxaparin, molecular dynamics

## Abstract

Apolipoprotein E4 (ApoE4) is thought to increase the risk of developing Alzheimer’s disease. Several studies have shown that ApoE4-Amyloid β (Aβ) interactions can increment amyloid depositions in the brain and that this can be augmented at low pH values. On the other hand, experimental studies in transgenic mouse models have shown that treatment with enoxaparin significantly reduces cortical Aβ levels, as well as decreases the number of activated astrocytes around Aβ plaques. However, the interactions between enoxaparin and the ApoE4-Aβ proteins have been poorly explored. In this work, we combine molecular dynamics simulations, molecular docking, and binding free energy calculations to elucidate the molecular properties of the ApoE4-Aβ interactions and the competitive binding affinity of the enoxaparin on the ApoE4 binding sites. In addition, we investigated the effect of the environmental pH levels on those interactions. Our results showed that under different pH conditions, the closed form of the ApoE4 protein, in which the C-terminal domain folds into the protein, remains stabilized by a network of hydrogen bonds. This closed conformation allowed the generation of six different ApoE4-Aβ interaction sites, which were energetically favorable. Systems at pH5 and 6 showed the highest energetic affinity. The enoxaparin molecule was found to have a strong energetic affinity for ApoE4-interacting sites and thus can neutralize or disrupt ApoE4-Aβ complex formation.

## 1. Introduction

Alzheimer’s disease (AD) is a neurodegenerative disorder characterized by synapse loss leading to a progressive decline in cognitive functions. Some characteristic symptoms in AD are short-term memory impairment and problems with spatial orientation, language, logic, and reasoning [1]. AD is also associated with aging and affects more than 6.2 million people over the age of 65 in the United States alone and is expected to increase to nearly 13.8 million by 2060 [2]. Early detection of AD is a crucial step to develop interventions designed to delay or stop the progression of the disease. Currently, early diagnosis is the most effective strategy to attack this disease through different pharmacological treatments. For the AD diagnosis, numerous biomarker candidates are being investigated, from metabolic to structural and functional neuroimaging studies, including sensory measures, digital biomarkers, blood levels of target proteins, etc. [3]. Recent advances in non-invasive diagnostic trials have allowed developing blood tests for the quantification of amyloid-beta (Aβ) levels with diagnostic accuracy [4,5]. These blood-based diagnostic tests may prove to be robust, accessible, and potential biomarkers for the early detection of AD.

One of the pathological hallmarks of AD is the extracellular accumulation of soluble amyloid-β (Aβ) and of insoluble Aβ as plaques in the brain parenchyma and in cerebral artery walls in cerebral amyloid angiopathy (CAA) [6]. Dysfunction of enzymatic degradation mechanism in the elimination of the Aβ from the brain and arteries leads to accumulation of plaques and that rise with increasing age and in AD [7]. Interestingly, apolipoprotein E (ApoE) is thought to be involved in Aβ plaque formation. Pathological evidence showed that in postmortem AD brain tissues, the ApoE4 exacerbates the intraneuronal accumulation of Aβ plaque deposition in the brain parenchyma [8], and promoted formation of neurotoxic Aβ oligomers/fibrilization [9,10]. Individuals afflicted with AD carrying the ApoE gene markedly increase the risk of late-onset Alzheimer’s disease (AD) [11,12]. Among them, the ApoE4 allele increases the risk of AD when compared to ApoE2 and ApoE3 carriers by showing a greater number of Aβ plaque deposition in brain parenchyma [13,14,15].

ApoE is an α-helical glycoprotein that in its mature form has 299 residues and it is produced in the liver and by almost all brain cells: vascular smooth muscle cells, astrocytes, microglia, choroid plexus, and neurons [16,17]. ApoE plays a key role in a wide variety of physiological processes, such as lipid transport or maintenance and neuron repair [18]. There are three isoforms of human ApoE that only differ in residues at positions 112 and 158, ApoE2 (both cysteines), ApoE3 (C112, R158), and ApoE4 (both arginines). Three major domains have been identified in ApoE structures, the N-terminal (residues 1–167) and the C-terminal (residues 206–299), linked by a flexible hinge domain (residues 168–205) [19]. The protein-protein interaction between ApoE4 and Aβ is not well understood and the role of ApoE in the deposition or removal of Aβ plaques is still unknown. Some studies suggest that ApoE interacts with Aβ in the receptor-binding region (residues R136–R150) as well as in the lipid-binding region (residues E244–M272) [20]. It is noteworthy that ApoE also contains two heparin-binding sites, residues R142–R147 that overlap with the Aβ binding region in the N-terminal domain, and residue K233 in the C-terminal domain. Likewise, Aβ contains a binding region (residues H13–L17) that can interact with both ApoE and heparin [21]. However, an in vitro and in vivo study reported that ApoE-Aβ interactions are minimal in physiological fluids [22]. This study proposes that ApoE influences Aβ metabolism through its competitive interactions with other receptors/transporters, e.g., low-density lipoprotein receptor-related protein 1 (LRP1). There are several studies in this regard and, in particular, some suggest increasing ApoE4 lipidation to reduce the intraneuronal accumulation of Aβ and thus alleviate cognitive impairment in ApoE4 targeted replacement (TR) mice [23,24]. Like LRP1, heparan sulfate proteoglycans (HSPGs) play an important role in neuronal interaction with Aβ, which also has a heparin-binding region [25]. HSPGs regulate and control the uptake of various cell surface proteins, e.g., tau, α-synuclein, and soluble amyloid precursor protein (APP) [26,27]. In this regard, it has been shown that enoxaparin, a low molecular weight glycosaminoglycan (GAG) form of heparin, can decrease cortical Aβ concentration, reduce the number of activated astrocytes around Aβ plaques and improve cognitive functions in AD transgenic mice [28,29,30].

Aβ peptides are the integral component of senile plaques in AD and recent reports suggest that Aβ oligomerizes and accumulates in endo-lysosomal vesicles at low pH [31]. Recent studies revealed indeed that low pH is found in the human brain and in the cerebrospinal fluid (CSF) from post-mortem AD brains compared to normal controls [32]. Acidic pH was reported to promote the self-assembly of Aβ oligomers, which may be a prerequisite for their neuropathogenicity, and their aggregation behavior in neuronal cells [33]. All this may be crucial in understanding the neurodegenerative process of AD. In vitro studies demonstrated that at pH 5.38, fibril aggregation of the Aβ 1–42 complex is promoted, and in PC12 cells apoptosis is induced [33]. In addition, in vivo studies showed that low pHs at the CSF increased Aβ plaque-load in APP/PS1 transgenic mice [32]. Taken together, these studies suggest the importance of pH in the brain microenvironment and how this may affect Aβ peptide aggregation in AD.

In this study, we aimed to elucidate the detailed molecular mapping of the ApoE4-Aβ complex interaction and the competitive antagonist effect of the enoxaparin on such complex, through molecular dynamics simulations (MD). Furthermore, we explored the effect of different pH conditions on the stability of the ApoE4-Aβ, complex and how the enoxaparin could neutralize those interactions. With the improved functionality and computational efficiency of molecular simulations, here we provide new insights into the conformational changes and stability of the aforementioned molecular complexes. These last two elements may have important therapeutic implications, both for the study of relevant specific binding sites and for the design of drugs related to AD.

## 2. Computational Details

### 2.1. System Preparation

Before studying the ApoE4-ligand complexes, all structures were stabilized by computational simulations. MD calculations were performed for the ApoE4 protein and Aβ peptide, and QM calculations for the enoxaparin molecule. To simulate the different pH conditions, only ApoE4 and Aβ structures were modified by performing a standard pKa calculation using the PROPKA 3.1 methodology implemented in the APBS-PDB2PQR server (Appendix A) [34,35,36]. Based on these pKa values and the pH to analyze, different protonation states for the ionizable residues were assigned using the *pdb2gmx* tool. For this study, pH values of 7.0, 6.0, and 5.0 were chosen.

#### 2.1.1. ApoE4 and Aβ Structures

To build the full-length human ApoE4 structure, the FASTA sequence of ApoE4 containing 317 amino acid residues was retrieved from the UniProtKB database (accession code P02649.1) [37]. To obtain the three-dimensional model, two protein structures of human ApoE were selected as templates: an ApoE4 protein fragment (PDB ID: 1B68 [38]), and the NMR structure of full-length ApoE3 (PDB ID: 2L7B) [19]. The first structure was chosen because it was used in interaction studies between the N-terminal domain with an enzyme-prepared heparin oligosaccharide. The second structure is a monomeric mutant of the ApoE3 protein with five mutations in the C-terminal domain (residues 257, 264, 269, 279, and 287) (Appendix A). The I-TASSER server was used to construct the ApoE4 structure and the best model was selected based on the C-score (−0.16, Appendix A) [39,40,41].

To analyze peptide Aβ, solution structure PDB ID: 1IYT was retrieved from the RCSB protein data bank [42]. This structure was chosen for showing two helical regions connected by a β-turn. Several studies mention the relevance of the helical conformation in this peptide since it could be related to its neurotoxicity [43,44,45]. In addition, peptide Aβ was obtained in apolar solutions, which makes it a good model to analyze the effect of pH on its structure.

To avoid molecular overlaps and maximum energy states, the ApoE4 and Aβ structures were optimized using the ModRefiner server [46]. The ModRefiner algorithm performs an adjustment of the atomic positions by refinement of the structure of high-resolution proteins. The quality and accuracy of this model were validated using the PDBsum server (Appendix A) [47].

#### 2.1.2. Enoxaparin Molecule

To obtain the molecular complexes with ApoE4, the enoxaparin structure was used in the docking calculations (PubChem CID: 772). Enoxaparin (Enx) is a low molecular weight synthetic heparin with a mechanism of action similar to that of heparin [26,48]. The structure was built with GaussView v.6 software package [49] and optimized by DFT calculations with Gaussian 16 software package [50] (Figure 1a). The optimization process was performed using the exchange-correlation functional CAM-B3LYP [51] and the TZVP base set [52]. The vibrational frequencies were calculated to ensure that the geometries were those of minimum energy. To investigate the electrostatic effect of this molecule as an ApoE4 ligand, atomic charges were calculated using the Hirshfeld population analysis [53,54,55] with implicit solvent effect (*SCRF = (SMD, Solvent = Water surfaces*)). The molecular electrostatic potential (ESP) surface was calculated to visualize the polar and non-polar regions of these ligands (Figure 1). The topologies and MD parameters of this molecule were obtained via the LigParGen server, [56,57,58] which uses the OPLS-AA force field parameters to generate them [59,60]. These topologies were reparameterized using the optimized structures and atomic charges obtained in previous quantum calculations.

#### 2.1.3. Molecular Docking between ApoE4 and Ligands

To study the binding interactions and obtain the initial structures of the ApoE4-ligand complexes, molecular docking calculations were performed. Molecular couplings were carried out in two steps. First, docking between ApoE4 and Aβ was performed to identify the major regions of interaction on the surface of ApoE4. In order to accomplish this, the PatchDock docking server was used to obtain the structure complexes based on complementary forms [61,62]. The results were refined on the FireDock server, allowing solutions to be re-scored based on an energy function. The use of the FireDock algorithm allowed to give flexibility to the system for optimal interaction of Aβ with ApoE4 [63,64]. For each pH studied, the top ten ApoE4-Aβ complexes were selected for further MD simulation studies. Second, once the main interaction sites were identified, enoxaparin molecules were placed in these regions using the AutoDock Vina program [65]. Ten ApoE4-Enx complexes were obtained for each interaction site and the complex with the highest score was chosen for the MD simulations. Both molecular complexes, ApoE4-Aβ and ApoE4-Enx were compared to know if enoxaparin was able to compete energetically with Aβ, thus neutralizing the interaction sites on the ApoE4 surface.

### 2.2. Md Simulations

All MD simulations were performed with GROMACS 2020.3 [66,67], and with OPLS-AA force field parameters. The protein systems were located in the center of a parallelepiped box in which the distance from the borders to the protein edges in all directions was 1.1 nm. These structures were solvated using the TIP4P [68] water model. To neutralize the systems and mimic physiological conditions, Na${}^{+}$ or Cl${}^{-}$ were added to obtain an ionic strength of 150 mM with a neutral total net charge. To avoid steric clashes and poor contacts, energy minimization was carried out using the steepest descent algorithm until reaching the maximum force of 500 kJ·mol${}^{-1}\cdot$ nm${}^{-1}$. To reach the desired temperature and pressure (309.65 K and 1 bar), equilibrium MD simulations were performed in the NVT and NPT ensembles with position restraints on heavy atoms. The modified Berendsen thermostat (V-rescaling algorithm) [69,70] and the Parrinello-Rahman barostat [71] were used for these purposes. The coupling constants values were fixed at $\tau_{T}=0.1$ ps, and $\tau_{P}=2.0$ ps. All systems were specified in periodic boundary conditions (PBC) in all directions (x, y, z). All simulations were carried out with a short-range unbound boundary of 1.2 nm. The Particle Mesh Ewald (PME) [72] method was used to calculate long-range electrostatic interactions with a tolerance of 1 $\times \;10^{5}$ for the contribution in real space. The Verlet neighbor search slicing scheme was applied with a neighbor list update frequency of 10 steps (20 fs). Bonds involving hydrogen atoms are constrained by the linear constraint solver (LINCS) algorithm [73,74]. Finally, the MD production simulations were carried out for 500 ns at 309.65 K and 1 bar, using the NPT ensemble without backbone constraints, and with an integration time step of 1 fs. For analyses, all trajectories were saved every 15 ps.

### 2.3. MM/PBSA Calculations

To assess the binding affinities of ApoE4-ligand interactions, molecular mechanics Poisson-Boltzmann surface area (MM/PBSA) calculations [75] were performed. This was done using the program g_mmpbsa [76], which calculates the binding energy components, with the exception of the entropic term, through an energy decomposition scheme. Although the g_mmpbsa methodology, as we have already said, does not include the calculation of the entropic term and therefore cannot calculate absolute binding free energies (BFE), it does calculate relative BFE. Therefore, this tool was used to compare the affinity of the different ways in which ligands bind to the same receptor sites. Free energy calculations and energy contributions per residue were performed to locate the main residue interactions and evaluate the effect of each residue on the ApoE-ligand complexes. MD simulations of 100 ns were performed and trajectories were analyzed to estimate the binding free energy (ΔG${}_{bind}$), which was calculated by the following equation:(1)$$\Delta G_{bind}=G_{complex}-\left(G_{ApoE}+G_{lig}\right)=\Delta E_{MM}+\Delta G_{sol}-T\Delta S$$
where $G_{complex}$ is the total free energy of the ApoE-ligand complexes. $G_{ApoE}$ and $G_{lig}$ are the free energies of the isolated ApoE structure and Aβ or Enx in the solvent, respectively. $\Delta E_{MM}$, represents the energy contributions of molecular mechanics. $\Delta G_{sol}$ is the solvation free energy required to transfer a solute from vacuum to solvent. The term *T*Δ*S* refers to the entropic contribution and was not included in this calculation due to computational costs [76,77,78]. Individual terms $E_{MM}$ and $G_{sol}$ were calculated as follows:(2)$$E_{MM}=E_{bonded}+E_{vdW}+E_{elec}$$
(3)$$G_{sol}=G_{p}+G_{np}=G_{p}+\gamma A$$

In Equation (Equation 2), the linked interactions are represented by the term $E_{bonded}$, and in the single path approach, $\Delta E_{bonded}$ is taken to be zero [64]. Unbound interactions are represented by the terms $E_{vdW}$ and $E_{elec}$. In Equation (Equation 3), the solvation free energy ($G_{sol}$) is the sum of the polar ($G_{p}$) and non-polar ($G_{np}$) contributions. The $G_{p}$ term is calculated by solving the Poisson-Boltzmann equation. For the $G_{np}$ term, the non-polar solvent accessible surface area (SASA) model was used, where γ (0.0226778 kJ/mol·A${}^{2}$) is a coefficient related to the surface tension of the solvent, and A is the SASA value. To ensure convergence of our MM/PBSA results, we have considered only the last stable 50 ns (last 250 frames) of the MD trajectories and evaluated them using FEL analysis for each complex. Frames were selected at a regular interval of 0.2 ns for better structure-function correlation. In addition, we use bootstrap analysis to calculate the average binding energy that is included in the g_mmpbsa tools. All calculations were obtained at 309.65 K and default parameters were used to calculate the molecular mechanical potential energy and the free energy of solvation [76]. Finally, the binding free energy by residue was obtained using:(4)$$\Delta G_{bind}^{res}=\Delta E_{MM}^{res}+G_{p}^{res}+G_{np}^{res}$$

### 2.4. Structure and Data Analysis

Statistical results, root mean square deviation (RMSD), root mean square fluctuation (RMSF), radius of gyration (RG), SASA, hydrogen bonds, free energies, structures, trajectories, and B-factor maps were obtained using Gromacs modules. The different analyses of the structural properties were performed using the MD trajectories of the last 200 ns for each isolated protein and the last 50 ns for the ApoE4-ligand complexes. Results were visualized using the Visual Molecular Dynamics (VMD) [79] software, UCSF Chimera v.1.14 [80], and Pymol v.2 [81]. Graphics were plotted using the XMGrace software [82]. 2D representations of electrostatic and hydrophobic interactions were constructed using the program LigPlot [83]. Electrostatic potential surfaces within the molecular mechanics framework were calculated in APBS (Adaptive Poisson Boltzmann Surface) software v.1.4.1 [84]. The pqr files were created in the PDB2PQR server [85]. Free energy landscape maps (FEL) were used to visualize the energy associated with the protein conformation of the different models during the MD simulations. These maps are usually represented by two variables related to the atomic position and one energy variable, typically the Gibbs free energy. The FEL maps were plotted using the sham gmx module with the RMSD and RG as the atomic position variables with respect to their average structure. Figures related to these maps were constructed using Wolfram Mathematica 12.1 [86].

## 3. Results and Discussion

Several studies have suggested that ApoE4-Aβ interactions are to be related to increased Aβ deposition in the brains and play an important role in Aβ senile formation plaques as well as with neurofibrillary tangles [87,88]. In addition, in vitro studies show that the ApoE protein forms complexes with Aβ through regions within their heparin-binding sites [21]. However, these types of interactions are still poorly understood. On the other hand, the use of heparin as a multitarget drug for Alzheimer’s disease is widely documented, including its interactions with the ApoE4 protein. However, its use is restricted by its powerful anticoagulant activity and, being a mixture of polysaccharides, it is difficult to find effective therapeutic doses. Therefore, in this work, enoxaparin was used as a therapeutic agent since it has been shown that it can be tolerated without side effects [48].

To better understand the effect of pH on ApoE4 interactions, several MD simulations were carried out under 3 different pH conditions: 7.0, 6.0, and 5.0, at 309.65 K. The 42 residues of the peptide Aβ and the enoxaparin molecule were used as ApoE4 ligands for these purposes.

### 3.1. Enoxaparin (Enx) Structure

Prior to MD simulations, the Enx structure was optimized with QM calculations to obtain the Hirshfeld atomic charges that would be used in the force field of this ligand. Enx is considered a small molecule ( 1.13 kDa) with a topological polar surface area of 652 Å${}^{2}$, due to sulfate and carboxyl groups that confer a electrostatic character [89]. Figure 1A shows the 2D structure and the quantum electrostatic potential (ESP) surface of Enx. A high concentration of negative charge can be observed in a large part of the molecule and a small area with a positive charge on nitrogen atoms present in the structure. Once the atomic charges were obtained, they were added to the OPLS/AA force field to evaluate the electrostatic effect of the ligand. Figure 1B shows the mechanical ESP surface obtained with these charges, which preserves the electrostatic character of the quantum ESP surface. The main advantages of Hirshfeld atomic charges are not to overestimate electrostatic properties and to speed up MD calculations [90,91].

### 3.2. pH Effect on Isolated ApoE4 Structure

Using the NMR structure of ApoE3 as the main template, the ApoE4 was modeled obtaining the initial structure for the in silico studies. Three main domains were taken into account for the structural analysis: N-terminal domain (NT, residues 1–167), hinge domain (HR, residues 168–205), and C-terminal domain (CT, residues 206–299) (Figure 2A). An extra-domain was considered in the analysis, named A-domain, and its residues were numbered −17 to 0 (Met-17 to Ala0).

To determine the initial docking structures, 500 ns MD simulations of the isolated ApoE4 and Aβ proteins were carried out at different pH concentrations. To assess the stability of the ApoE4 systems, root mean square deviation (RMSD) was calculated of the protein atoms with respect to initial conformation. As shown in Figure 2B, all systems trend to an asymptotic curve after 300 ns. However, even though the pH6 structure has the lowest RMSD value (0.59±0.03 nm), a slightly positive slope and highest fluctuation can be seen, which seems to indicate that the system is not yet fully converged. On the other hand, pH7 and pH5 structures show a low fluctuation (0.61±0.01 and 0.67±0.01 nm, respectively), indicating that these systems reached equilibrium (Table 1).

An important parameter that can condition protein-ligand interactions is the exposed area of the receptor. To assess this parameter solvent-accessible surface area (SASA) of each system was computed for the entire MD trajectory (Figure 2C). All SASA values were significantly similar, being the structure at pH5 the one that showed the lowest value (168.78 ± 5.20 nm${}^{2}$). Furthermore, when the degree of systems compaction was analyzed, greater differences are observed between the structure at pH5 with respect to the others, obtaining lower values in the three calculated axes (Figure 2D and Table 1). These results seem to indicate that some compaction degree of the ApoE4 is carried out in the core of the proteins and in a similar way on the exposed surface.

When performing the analysis of the fluctuation per residue of the three structures, a region of high fluctuation can be observed in the RMSF diagram (Figure 3A). This region comprises residues G182-R215 and shows greater fluctuation in the structure at pH6, which would cause less compaction and its structural non-convergence. These high vibrations of the residues can be associated with increased activity and propensity to interact with other molecular systems [92,93]. When analyzing the B-factor on the molecular surfaces of the structures (Figure 3B), it is observed that in the structure at pH6, the residues with high fluctuation are distributed throughout the protein, while in the other structures, only small areas with high vibrations were observed.

#### ApoE4 Closed-Conformation Structure

The great stability of the ApoE4 protein shown in the MD calculations is reflected in the conservation of its globular shape despite changes in pH (Figure 4A). These results are in good agreement with experimental studies that mention that this protein is less susceptible to chemical and thermal denaturation than the ApoE3 and ApoE2 structures [18,94]. However, these studies also reported that this globular shape loses stability as the pH decreases to values close to 4. At low pHs, ApoE4 forms unfolded states characteristic of molten globules due to interactions between the NT and CT domains.

The interactions between the NT and CT domains in ApoE4 have already been investigated and it has been mentioned that they play an important role in the characterization of its structure [95,96]. To investigate possible reasons why the structures are highly conserved in the simulations, BFE and H-bond calculations between the NT and CT domains were performed. H-bond analysis showed that the structures are stabilized by a network of electrostatic interactions that exhibit significant occupancy along the trajectories. Especially residues R226, R228, E245, E255, R260, R274, and D297 present high H-bond occupancy values. The position and bonds between the residues can be seen in the Circos plots (Figure 4B). As can be seen from the figures, the CT domain is stabilized by numerous hydrogen bonds throughout this domain. As the pH becomes acidic, the number of interactions decreases, which is reflected in the number of residues with significant occupancy (Figure 4C). At pH6 and 7, there is a greater number of h-bonded residues between these domains, however, at pH7 the occupancy is more stable. It is important to mention that the A-domain does not interact with the CT domain and that the HR-domain does so with few residues.

When analyzing the BFE, it was observed that there are several residues that provide favorable energies for the stability of the closed form of ApoE4, which can be seen in the heat maps of the Circos plots. The results show that the region between residues L133 and R158 of the NT domain interacts with residues R224 to Q246 of the CT domain with low BFE values (<−200 kJ/mol). This region of interaction has been reported in a contact distance study [96]. Other residues contribute with strong energies to the closed form, R103, R114, in the NT domain; Y162, E171, R198, R207, R209, in the HR domain; and E255, R260, R274, E281, D297, H299, in the CT domain. All of the energies are below −150 kJ/mol. It is interesting to mention that when making a study of the location of epitopes in the structures (histogram plots), these epitopes are located at the areas of high fluctuation. As can be seen, the epitope regions are highly conserved in the different structures studied.

Finally, when performing the alignment with the three final structures of ApoE4 proteins (Figure 4D), it is observed that the structure is highly conserved under the different pH conditions. Only a slight loss of structure stability was observed in ApoE4 at pH5 (orange color). As mentioned above, this loss of stability and its new molecular conformation are in good agreement with experimental studies. The changes in the ApoE4 protein would be related to the partial opening of the structure due to the loss of alpha-helix structures, taking on a molten globule-like conformation [18].

### 3.3. pH Effect on Isolated Aβ Structure

As a major component of senile plaques, the Aβ-42 peptide is considered the most neurotoxic form due to its fast self-assembly [97,98]. This form is a byproduct of amyloid precursor protein (APP) proteolysis. APP is a membrane protein that serves as a regulator of neuronal plasticity and synapses [99,100,101]. For this work, the Aβ structure used in the MD calculations has the PDB-ID: 1iyt, and was obtained by NMR techniques in apolar solution. This structure has been used in various MD studies on conformational transitions of the Aβ monomer [102,103,104,105]. This Aβ model has two alpha-helices comprised between residues D7–S26 and G29–V39, joined by means of a β-turn type structure (Figure 5A). To simulate pH changes in the Aβ structure, the protonation state of three histidines (H6, H13, and H14) was changed based on PROPKA results (Appendix A). Three replicas were simulated to compare the folding behavior of the Aβ peptide (Appendix A) and to obtain the structure to be used in the interactions with ApoE4 at these low pHs. The results show that the Aβ loses its helical structures becoming a tangled disordered structure, which would be the first step to be able to form species that contain structures rich in β-sheets and so to self-assemble [106,107]. This disordered structure is manifested in all the stability indicators analyzed (Appendix A), in addition to presenting a high fluctuation in many of its residues (Appendix A). Moreover, in vitro studies showed that the preferential binding site of ApoE to the Aβ was carried out when Aβ was without α-helices structures [108]. For this reason, replica 1 was chosen for further analysis at low pH, besides being the structure with the largest surface area (SASA = 37.23 ± 2.27 nm${}^{2}$).

Similarly, when comparing the structure at pH5 with the one obtained at pH7, the results showed that both structures have a compact and globular shape in the MD simulations (Figure 5A). However, the structure at pH7 shows a rapid and greater convergence (1.38 ± 0.01 nm) with respect to the structure at pH5 (1.16 ± 0.13 nm), which is observed in the RMSD plot (Figure 5B). With a RMSF value of 0.64 ± 0.16 nm, the high instability of the structure at pH5 is due to the high fluctuation of the two α-helices during the folding process. The radius of gyration plot shows different bending behaviors in the structures. While the structure at pH7 reaches convergence at 100 ns in the MD, the structure at pH5 has high folding variations due to the loss of this secondary structure. Although with very similar mean values in intra- and intermolecular hydrogen bonds, a greater fluctuation in the formation of these bonds can be observed in the structure at pH5, which explains its low stability. These results suggest that a low pH favors the loss of alpha-helix structures, which would allow the possible formation of beta-sheets, necessary for their self-assembly [106,107].

When analyzing the minimum energy structures, it was found that both conformations lose the initial α-helices. Nevertheless, the structure at pH7 formed a new α-helix (D23–M35), which gives it some structural stability (Figure 5C). Another important characteristic is that the hydrogen bonds it forms are between opposite regions of the protein (D1–K28, E3–V39, R5–D23). On the other hand, the structure at pH5 loses its helical structure and maintains its compact shape due to locally formed hydrogen bonds.

The change in pH conditions also produces a change in the electrostatic properties of the structures. In the case of Aβ, it is observed that at pH7 there is a greater surface with nucleophilic characteristics, denoted by the red coloration of its surface, which generates a greater interaction with positively charged systems. On the other hand, the structure at pH5 loses this characteristic, increasing areas of an electrophilic character (blue coloration) and regions of neutral charge. These results are interesting since they seem to indicate that at low pHs, Aβ increases its ability to interact with both hydrophilic and hydrophobic systems, which increases its promiscuity.

### 3.4. ApoE4 Complexes

To assess the ability of Enx to inhibit Aβ interaction sites, ApoE4-Aβ molecular docking calculations were performed to obtain the sites with the highest probability of interaction at the different pHs studied. Although these sites have been extensively studied [17,21,25,109,110,111], these analyses were performed with incomplete or open ApoE4 structures, so this work explores new interaction sites. Once the sites were obtained (Figure 6), Enx was placed on these sites by local docking and 100 ns of MD simulations were performed to obtain the interaction BFE.

The analyses of the ApoE4-ligand interactions were performed only for those complexes that presented the highest BFEs in each of the interaction sites found at the pHs studied. Figures from the site labeled S1 are shown in the main text, while figures from the other interaction sites are attached in the Appendix A. The data and values of all simulated complexes are presented in the different tables of the manuscript.

#### Interaction Sites

Based on the top ten solutions of the ApoE4-Aβ complexes, several interaction sites were found in molecular docking analyses. At pH7, four interaction sites were found, labeled S1, S2, S3, and S4, respectively (Figure 6A). The S4 site was the one that most interacted with the Aβ solutions (1, 3, 4, 5, 7, and 8), being this region with the highest affinity in the formation of molecular complexes in the docking calculations. These structures show mainly hydrophobic interactions, however, it was at this site that the greatest number of electrostatic interactions occurred with six ApoE4 residues: T83, P84, T89, S263, W264, and D271 (Table 2). The S2 site is located between the A and NT domains. Two Aβ (solutions 6 and 9) were located in this region. At this site, there were also hydrogen bond type interactions with residues L-5, G-3, C-2, E13, R15, and T18.

The S1 and S3 sites presented only an Aβ structure and few electrostatic interactions, residue R215 for the S1 site and residues R142 and S296 for the S3 site. Experimentally, the regions between residues R136–R150 and E244–A272 of ApoE4 are recognized as the sites of interaction with Aβ [17]. These two regions are included in the S4 site, which is consistent with the experimental results.

For molecular docking at pH6, Aβ only interacted in three sites with ApoE4 and they were located in the same regions as at pH7, these sites were S1, S3, and S4. At these pH conditions, electrostatic interactions increased in all complexes, indicating a greater affinity of Aβ for ApoE4. At this pH, the S1 site was the one that formed the highest number of complexes with Aβ (solutions 1, 2, 4, 5, 7, 8, and 9) being residues K-16, W-13, R25, E27, R32, R206, A207, Q208, and R213, which form hydrogen bonds with Aβ. The S3 site only formed a complex with solution 10. The electrostatic interactions found were with residues R228, L229, D230, and T289. For the S4 site, two complexes were formed (solutions 6 and 9) and the electrostatically interacting residues were W20, T83, R167, E266, and P267.

Finally, at pH5 docking results show the formation of two new interaction sites (S5 and S6) and it is confirmed that the S1 and S4 sites are the ones with the highest probability of ApoE4-Aβ interaction. Under these conditions, the number of electrostatic interactions decreases in sites S1 and S4, with respect to the pH6 sites, which causes a lower affinity between the two proteins. Again, the S1 site is the one with the highest number of complexes formed (solutions 1, 2, 3, 4, and 10) with electrostatic interactions at residues K-16, L-5, E19, and Q24. The S4 site presents two complexes (solutions 5 and 7) with four hydrogen bonds at residues E7, V269, E270, and K282.

The S5 site has two molecular complexes, although there is a higher affinity of Aβ with this site as five electrostatic interactions are formed with residues R119, R180, R189, R191, and K242. Only a complex is formed with the S6 site, with residues Q46 and Q123 forming hydrogen bonds with Aβ.

### 3.5. ApoE4-Ligand Complexes after MD Calculations

Once the interaction sites were obtained, MD simulations were performed to obtain the BFEs of the ApoE4-ligand complexes. Stability indicators, mean values, and standard deviation for all ApoE4 ligand complexes can be seen in the Appendix A. After 100 ns of MD simulations, the RMSD plots show that most of the ApoE4-ligand complexes converge. However, in the Aβ complexes, solutions 2 (pH5) and 7 (pH6), both located at the S1 region, show a greater fluctuation due to the instability of the Aβ. These complexes present values of ± 0.20 and ± 0.18 for the standard deviation from solutions 2 and 7, respectively (residues 300–341 in RMSF plots). In particular, there are more intermolecular H-bonds between ApoE4 and Aβ in solutions at pH6 with a maximum of 9 ± 2, followed by solutions at pH5 with a maximum of 7 ± 2 and finally the solutions at pH7 with a maximum of 5 ± 1. These results suggest that there is a high structural affinity in ApoE4-Aβ complexes under low pH conditions.

On the other hand, Enx shows high stability in ApoE4 complexes, highlighting that the formation of intermolecular H-bonds at pH7 and 5 is similar to that observed in Aβ complexes, which implies similar structural affinities. At pH6, the lowest H-bonds formation is observed, ∼50% lower than that observed in Aβ complexes, so that under these conditions, Enx is structurally less competitive than Aβ.

### 3.6. Binding Free Energies (BFE) Analysis

Several studies refer to the high affinity of Aβ for ApoE [17,112]. Although the role of ApoE4-Aβ complexes is still unclear, experimental studies suggest that they are related to the damaging acceleration of Aβ aggregates in AD [17,113,114]. Low molecular weight heparin treatments have been shown to be effective in reducing Aβ concentration and deposition in transgenic mice [26,28,48]. In particular, Enx has been shown to be well tolerated by AD patients on long-term treatment [25].

To clarify whether Enx can compete energetically with Aβ at the found interaction sites and inhibit the formation of ApoE4-Aβ complexes, BFE calculations were performed to obtain their energetic affinities using MD simulations in NPT ensemble for 100 ns. All trajectories were saved every 0.2 ns and the last 50 ns were used to obtain BFE. The results for each pH can be seen in Table 3, Table 4 and Table 5.

Positive BFE values were obtained in the ApoE4-Aβ interactions under pH7 conditions. The highest BFE value was located at the S2 site (202 kJ/mol), and two more at the S4 site (Table 3). A high repulsive electrostatic energy was observed in these solutions due to unfavorable interactions between ApoE4 and Aβ residues. That was reflected in their intermolecular H-bonds, 3 ± 1 in the solutions at the S4 site, and 2 ± 1 at site S2 (Appendix A). The most favorable BFE was obtained in solution 2 (−388 kJ/mol) and was located at the S1 site. For sites S2, S3, and S4, the best energetic affinity was obtained in solutions 9 (−147 kJ/mol), 10 (−137 kJ/mol), and 4 (−230 kJ/mol), respectively. These solutions were used in the site interaction analyses, which will be discussed later.

In the case of the Enx complexes, all the interactions were energetically favorable at the different interaction sites, with the strongest affinity being at the S4 site (−405 kJ/mol). Under these conditions, Enx proved to be energetically competent at all interaction sites.

At pH6, most of the complexes bound to the S1 site show a high affinity for Aβ as seven solutions were located at this site (Table 4). The BFE obtained in these solutions ranged between −321 and −805 kJ/mol, being the site where more interaction complexes were observed in all the pH conditions studied. For this site, solution 1 (−805 kJ/mol) was chosen for further analysis. Taking the most favorable BFE values sites S3 and S4 increased their energetic affinities with respect to the results at pH7 by 300% and 200%, respectively. The complexes chosen at these sites were solutions 10 (−442 kJ/mol) and 3 (−406 kJ/mol) for sites S3 and S4, respectively.

Regarding the energies obtained in the ApoE4-Enx interactions, the affinity with the S1 site also strengthens, although to a lesser extent than with the Aβ. The same trend is repeated with site S3, where BFE gets more than 100% stronger. In contrast, for site S4, the BFE gets weaker (−146 kJ/mol), even though this interaction is almost twice the value compared to that of Sol6 of the Aβ complexes (−85 kJ/mol). These results show a higher affinity of Aβ with respect to Enx, which is complemented by the increase in the number of hydrogen bonds formed. The results suggest that at pH6, ApoE4-Aβ interactions are strong and difficult to neutralize.

At pH5, the strongest interaction energies between ApoE4 and Aβ were obtained. The results confirm that the S1 site is the one with the highest affinity for Aβ showing the highest BFE of all the complexes (−1106 kJ/mol). It can be seen that this strong BFE value was largely due to the contribution of electrostatic energy, showing the additional structural affinity between ApoE4 and Aβ residues (Table 5). Five solutions were tested at this site and all BFE values were below −300 kJ/mol. Solution 4 was the complex of choice for further analysis. For the S4 site, the energetic affinity increased, with solution 7 being the one with the best BFE (−709 kJ/mol). For the S5 and S6 sites, the chosen complexes were solutions 8 (−931 kJ/mol) and solution 9 (−164 kJ/mol). In the ApoE4-Enx complexes, the BFE was strengthened, improving the affinity shown at pH6. However, the BFE values were weaker than those obtained in the complexes with Aβ.

### 3.7. Intermolecular Contact and BFE Analyses

To elucidate which residues favor the interactions between ApoE4 and the Aβ and Enx ligands, an analysis of the different contributions per residue in each of the interaction sites found was performed. For this, the complexes that presented the most favorable energetic affinity were used and their minimal structures were analyzed to obtain the electrostatic and hydrophobic interactions. The figures corresponding to sites S2–S6 and all the information about the residue contacts are found in the Appendix A.

#### 3.7.1. S1 Site

The site labeled as S1 was the one that showed the highest affinity, both structural and energetic, in the molecular complexes formed by ApoE4 and Aβ. This site is located between domains A(M-17–C-2), NT (residues P12–E70), and CT (residues R206–M218) (Table 2).

Figure 7 shows the different ApoE4-ligand complexes formed in the S1 site at the different pHs analyzed, as well as their main electrostatic interactions. The contact analysis showed that the complexes at pH6 were the ones that had the greatest interaction along the MD trajectories. In the Aβ complex, 19 residues of ApoE4 interacted with 18 of Aβ (pH7, 13 and 11; pH5, 14 and 13), while 13 ApoE4 residues interacted with Enx (pH7 and pH5, have 11 residues). The numerous interactions observed in the ApoE4-Enx complex are due mainly to the Enx molecular size (652 Å${}^{2}$) and its electrostatic characteristics (Figure 1B). These features allow Enx greater penetration within the protein structure and also good affinity, both energetically and structurally. On the other hand, analyses on the calculations of the BFE distribution indicated that the most favorable interaction energies for the Aβ complex occurred at pH7 (Table 6). However, there were a large number of residues with strong energies that neutralized the BFE of the complex (Figure 8). It is interesting to mention that at pH7, the ApoE4 residues showed the strongest contributions to total BFE in the interactions with Aβ, thus the stability of the complex was mainly due to the structure of ApoE4. Whereas, at low pHs, the energy contributions of Aβ residues increase, indicating a higher affinity of ApoE4 for Aβ. The complex at pH5 was the one that presented the highest energy affinity of all the complexes studied. Both structures at this pH showed high interaction energies, thus increasing the stability of their respective complexes. The same trend is observed in complexes with Enx, in which the affinity of ApoE4 for Enx increases as the pH of the system decreases.

In addition, several regions with energetically favorable interactions can be observed in the molecular complexes (Figure 8). On the ApoE4 structure, within the ApoE4-Aβ complexes, those regions are M-17–L-5, P12–R38, R61–K75, R206–R226, and V269–M272; and on the Aβ structure, R5–D23, and A30–I32. Then, a region with a lower energy contribution is also identified, M-17–L-5, R15–D35, E49–L51, and R213–A216 for the ApoE4-Enx complexes.

#### 3.7.2. BFE Contribution on the Interaction Sites

Figure 9 shows the different contributions per residue to the total BFE of the analyzed complexes. This analysis was performed considering the different interaction sites found for each pH studied. At pH7, the results show that the energy contribution of the residues fluctuates greatly in the structure of ApoE4 in the different complexes. Especially the residues E66 (−114 kJ/mol), E70 (−117 kJ/mol), in S1 site; E219 (−132 kJ/mol), E220 (−119 kJ/mol) and D227 (−138 kJ/mol) in S2 site; E3 (−97 kJ/mol), E7 (−96 kJ/mol) and E281 (−107 kJ/mol), in S3 site; and E168 (−99 kJ/mol) in S4 site. The acidic residues E and D were the ones that generated the greatest repulsion to interactions with Aβ. Appendix A shows the main electrostatic interactions of the complexes formed at this pH. Comparing these residues to their BFE (Appendix A), there are several that have significant contributions to total BFE, M-17, K-16, and L-5 in S2 site; K233 in S3 site and R167, R90, and W20 in S4 site.

In the case of Enx, the energy contributions are well localized, being residue K157 (−87 kJ/mol), the residue that contributed the most of the BFE to the S4 site. Furthermore, this residue binds sequentially to R158 (−21 kJ/mol), which is distinctive of ApoE isoform 4. It is important to mention that at this pH the strongest BFEs of the four interaction sites were considered. This result is remarkable because the receptor-binding region (residues 136–150), the lipid-binding region (residues 244–272), and the heparin-binding sites are located at this S4 site [17], which would reinforce the proposal of this work.

For the complexes at pH6 and 5, it can be seen that there are no large repulsion energies between the structures of ApoE4 with Aβ. Furthermore, the residues that contribute significantly to the total BFE are located in regions of high interaction. At pH6, residue analysis shows that at the S4 site, Enx is energetically competitive with Aβ. The BFEs of the residues with the highest binding energy contribution in both complexes (W20, −160 kJ/mol for Enx, and R167, −162 kJ/mol for Aβ), are comparable and of similar magnitude (Appendix A). If residues W20 and E19 are taken from both complexes the BFE is even stronger. This can be seen in Figure 9C,D. It is noteworthy that several Aβ residues have more favorable energies than those obtained in ApoE4: Y10 (−160 kJ/mol), H14 (−191 kJ/mol), which means an important contribution to the stability of the complex (Table 6). It is also important to mention that H14 was one of the residues that changed its protonation state in the PROPKA analyses. This change could be the reason that its energy contribution was significant in the total BFE.

The residues that contributed the most to the total BFE, for the Aβ were, R25, L28, and W210 for the S1 site; E234, and H299 for the S3 site; and W20, R167, E266, and P267 for the S4 (Table 6 and Appendix A). For the Enx complexes, D35, and E50 for the S1 site; R251, and V294–N298 for the S3 site; and E19, and W20 for the S4 site. At this pH, the strongest interaction energies were obtained in the Enx complexes, showing that under these conditions Enx has a higher affinity for ApoE4 than at the other pHs.

Finally, pH5 is where the strongest interaction energies between ApoE4 and Aβ were obtained, especially at sites S4 and S5, the latter with a higher number of H-bond interactions (Figure 9 and Appendix A). The strongest BFEs per residue occur at sites S1 (K-16, and E19), S4 (L261, and K262), and S5 (R61, R189, and R191) (Appendix A). The S4 site has the highest number of residues with favorable BFE. These results show that S4 is the site of the ApoE4 structure that has the highest affinity for Aβ at the pHs studied. Aβ also exhibits strong interactions with ApoE4, especially at the S5 site, with three residues contributing to complex stability at this site, E22 (−234 kJ/mol), E3 (−213 kJ/mol), and D23 (−143 kJ/mol). The D23 residue also contributes strongly to the other interaction sites.

Although with lower energy, Enx also increased its interaction with ApoE4, having the highest affinity at the S6 site and especially with residues E45 (−125 kJ/mol), and Q48 (−104 kJ/mol). Strong interactions with residues V6 (−101 kJ/mol), and E7 (−64 kJ/mol) were also present at the S4 site (Appendix A), showing that Enx can also compete with Aβ at this site.

### 3.8. Study Limitations

Several study limitations and modeling assumptions may have affected our results: Firstly, the effect of pH was carried out only by changing the protonation states of some ionizable residues as a function of pKa values, at the beginning of the MD simulations. Currently, many methodologies allow to work under constant pH and also to change the protonation states during the simulations, however, Gromacs does not support the latter type of calculation. Nevertheless, an analysis of pKa values of histidines in Aβ structure shows the importance in the pH studies of the protonation states. As seen in Appendix A, pKa values of H6 and H14 remain almost constant throughout the MD simulation. On the other hand, histidine 13 has a remarkable change in its pKa, going from a value of 7.14 to a minimum value of 5.38. These differences are very important when assessing the protein interactions.

Secondly, the enoxaparin molecule has ionizable groups that can become protonated/ deprotonated when the environment changes. For all the pHs analyzed in this work, these groups remained in the protonated state (neutral species), which could affect the results obtained in the ApoE4-Enx interactions. We justify the use of this structure since enoxaparin had several sulfates and carboxyl groups and the non-convergence of our quantum calculations prevented us from using an ionized structure of this molecule.

Finally, enoxaparin is considered a small molecule compared to the unfractionated heparin. Although they share similar physicochemical and pharmacological characteristics, heparin probably has different ways of interacting with proteins.

Despite all these limitations, this study shows promise in understanding the effect of pH on AD pathogenesis and opens alternative avenues to explore new therapies through in silico analyses. Nonetheless, future studies should focus on understanding the role of ionizable groups in ApoE4 and Aβ interactions.

## 4. Conclusions

Several studies have shown that ApoE4-Aβ interactions may have an impact on the neuropathology and progression of the Alzheimer’s disease. However, the relationship in the way they interact is still unknown. Experimental studies have pointed at two interaction sites between the ApoE4 and Aβ peptides between residues R136–R150 and E244–A272. However, these interactions sites have been identified in experimental conditions that mimicked the physiological pH or under conditions where the brain is already damaged (post-mortem). Our study shows the effect that the pH has on the interactions between the Aβ and Enx with the ApoE4 protein by an in-depth in silico analysis. Here we show that the closed structure of ApoE4 protein can be stabilized by a network of H bonds that keeps the CT domain in union with the other domains. Remarkably low pH conditions increase the binding affinity of the Aβ for the ApoE4 protein. This increase in binding affinity was explained by the creation of new binding sites on the ApoE4 which showed higher energetic affinity compared with physiological pH conditions. In particular, the site identified in the S4 domain of the ApoE4 had the highest affinity for the Aβ even at the different pH conditions used in this study. This site contains the major experimentally determined ApoE4 interaction domains and strong energetic affinities, mainly under pH5 conditions.

Enoxaparin was found to have a strong competitive binding affinity for the S4 domain in the ApoE4 protein thus acting as an antagonist and ultimately blocking the formation of the ApoE4-Aβ complex.

These findings indicate the therapeutic potential of Enoxaparin in the treatment of Alzheimer’s disease.

## Figures and Tables

**Figure 1 biomolecules-12-00499-f001:**
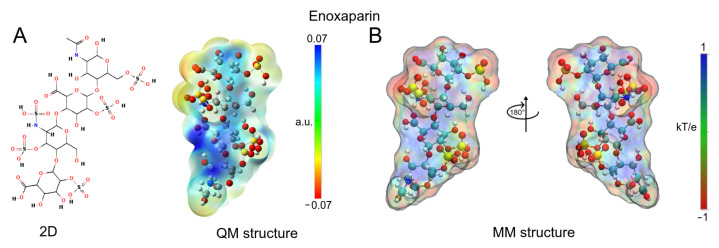
QM and MM of enoxaparin (Enx) structures. (**A**) 2D and 3D depiction of Enx. The optimized structure and ESP surfaces were obtained by DFT calculations. (**B**) ESP surfaces were obtained with APBS methodology and Hirshfeld atomic charges. On all surfaces, the different colors indicate their molecular electrostatic properties; red for the most nucleophilic zones; dark blue for the most electrophilic zones, and green for neutral.

**Figure 2 biomolecules-12-00499-f002:**
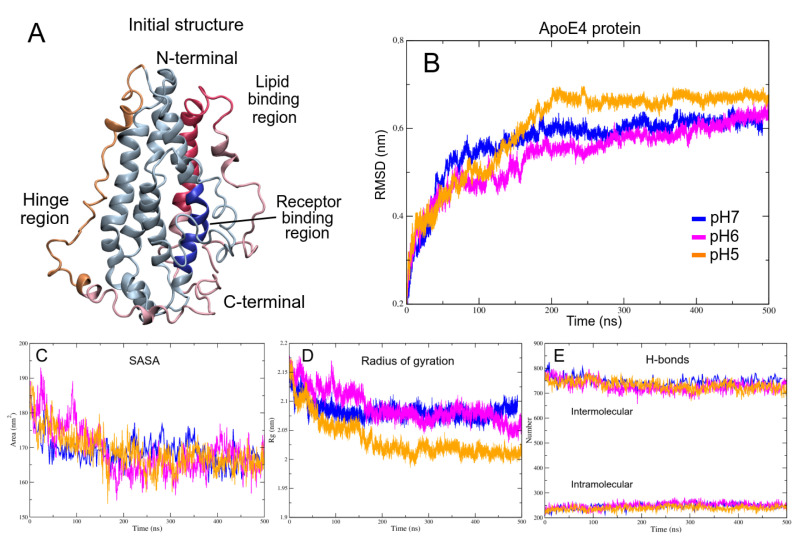
The initial structure of ApoE4 and its stability indicators. (**A**) Main regions of full-ApoE4 structure. Blue color corresponds to N-terminal; yellow color to Hinge region and red color to C-terminal region. (**B**) Root mean square deviation. (**C**) Solvent accessible surface area. (**D**) Radius of gyration. (**E**) Hydrogen bonds.

**Figure 3 biomolecules-12-00499-f003:**
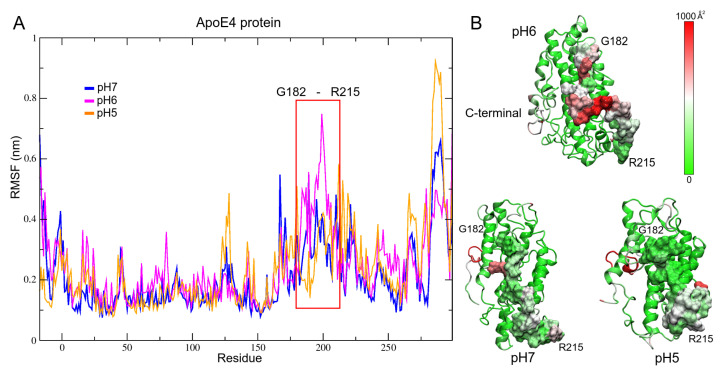
Fluctuation analysis of the ApoE4 structures. (**A**) RMSF plot of the last 300 ns of the MD trajectories. The largest fluctuation in the structures is shown in the red box. (**B**) B-factor mapped onto ApoE4 structures at different pH. The marked area corresponds to the region of greatest fluctuation. Green, white, and red colors indicate low, intermediate, and high fluctuations, respectively.

**Figure 4 biomolecules-12-00499-f004:**
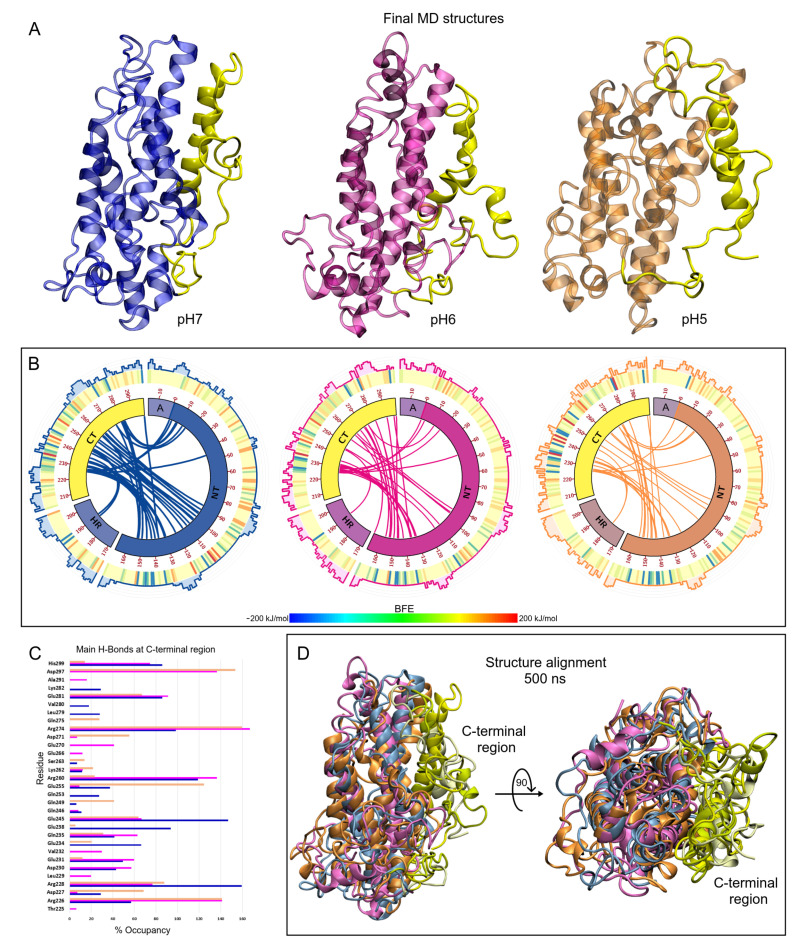
Final structures of the ApoE4 after 500 ns MD simulations. (**A**) Minimum energy structures of ApoE4 under different pH conditions. (**B**) Circos diagrams of the full-length ApoE4 structure. The main domains are represented by bands and the internal colored lines indicate hydrogen bonds between residues. Orange, magenta, and blue colors represent pH5, 6, and 7 conditions, respectively. The outer graphs represent the BFE heat map and the epitope probabilities for each residue. On the heat maps, blue, yellow, and red colors indicate favorable, neutral, and unfavorable BFE, respectively. In the epitope plots, the same color scheme was used to represent the different pH conditions. (**C**) H-bond occupancy formed between C-terminal residues and the remaining residues of the ApoE4 structure along the MD trajectory. Color bars indicate different H-bonds for each residue. (**D**) Structural alignment of ApoE4-Aβ complexes at different pHs after 500 ns of MD calculations.

**Figure 5 biomolecules-12-00499-f005:**
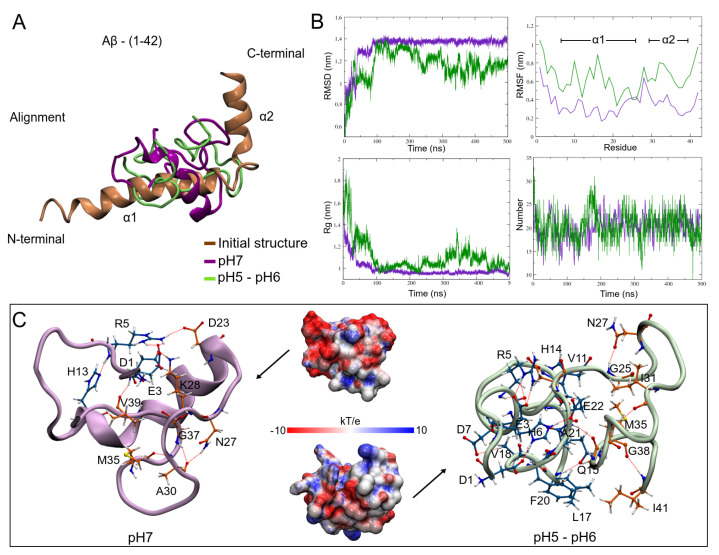
Results of the MD simulations on the Aβ structure. (**A**) Alignment of the initial structure with the minimum energy structures obtained at the studied pHs. (**B**) Stability indicators for the analyzed systems. (**C**) Network of hydrogen bonds in the compact structures of the Aβ and its effect on the electrostatic properties of the systems. The blue color indicates electrophilic regions, the color red, nucleophilic regions, and the white, neutral regions.

**Figure 6 biomolecules-12-00499-f006:**
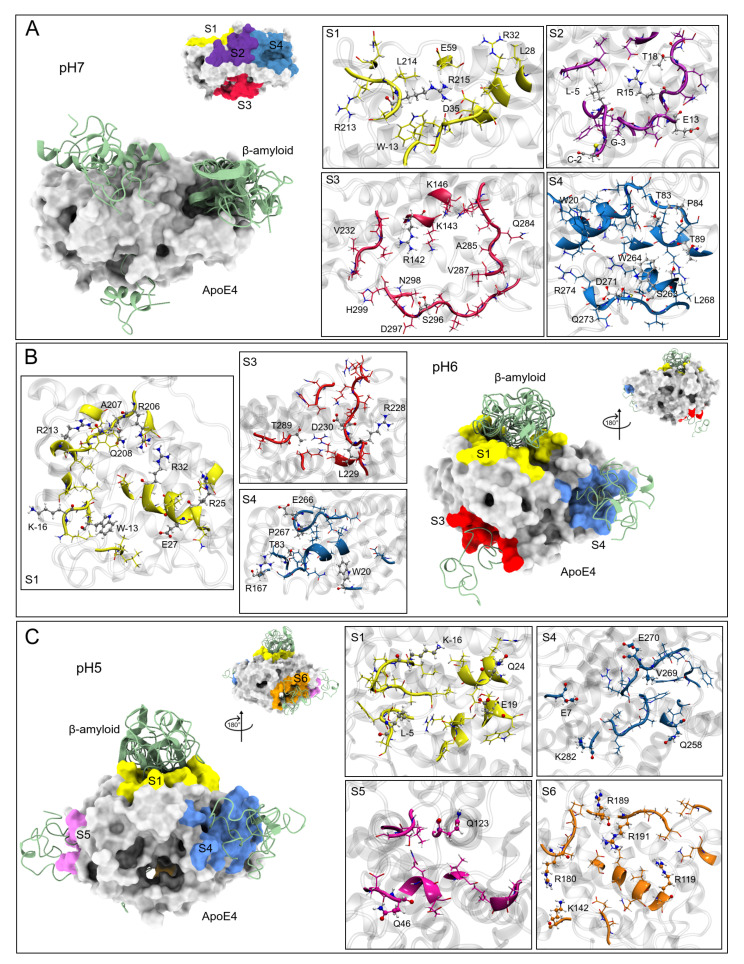
ApoE4 and Aβ interaction sites at different pHs obtained by molecular docking calculations. (**A**) Structures at pH7 and interaction sites found. (**B**) Sites of interaction at pH6. (**C**) Sites of interaction at pH5. In all figures, peptide Aβ is in green, and the surface of the ApoE4 structure is in gray. The colors of the interaction sites are the same for all pHs, S1 (yellow), S2 (purple), S3 (red), S4 (blue), S5 (orange), and S6 (magenta).

**Figure 7 biomolecules-12-00499-f007:**
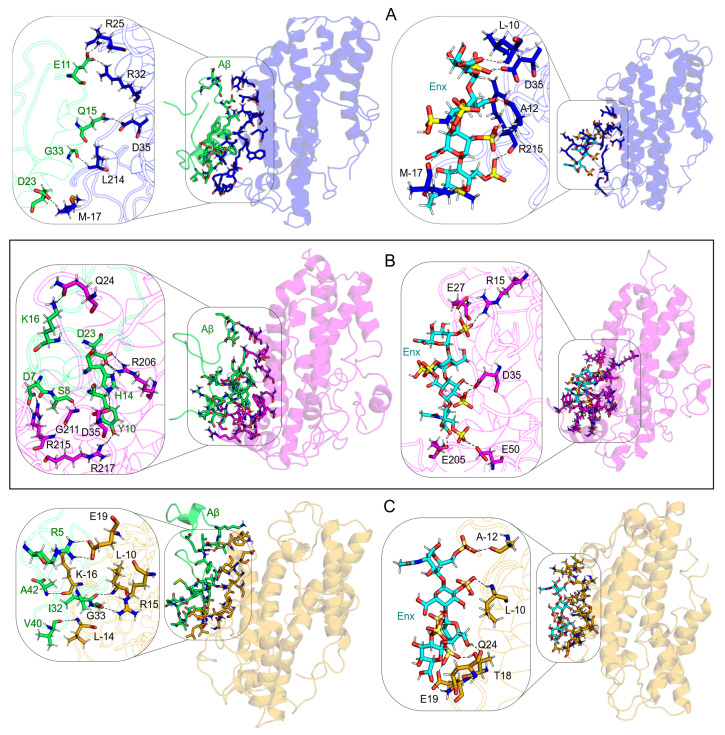
H-bond interactions at S1 site in ApoE4-Aβ-Enx complexes. (**A**) At pH7; (**B**) at pH6; and (**C**) at pH5. For ApoE4 residues, the colors blue, magenta, and orange were used to represent pH conditions, respectively. For all pH values, the Aβ residues were colored green and the Enx molecule cyan.

**Figure 8 biomolecules-12-00499-f008:**
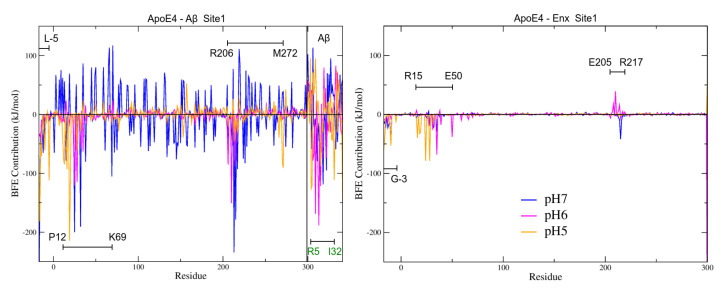
MM-PBSA calculation of BFE per residues in the S1 interaction site at different pHs. The left panel shows the ApoE4 residues with the strongest binding energies with Aβ as ligand. The right panel with Enx as ligand. The same color code was used for both panels to represent the different pH conditions.

**Figure 9 biomolecules-12-00499-f009:**
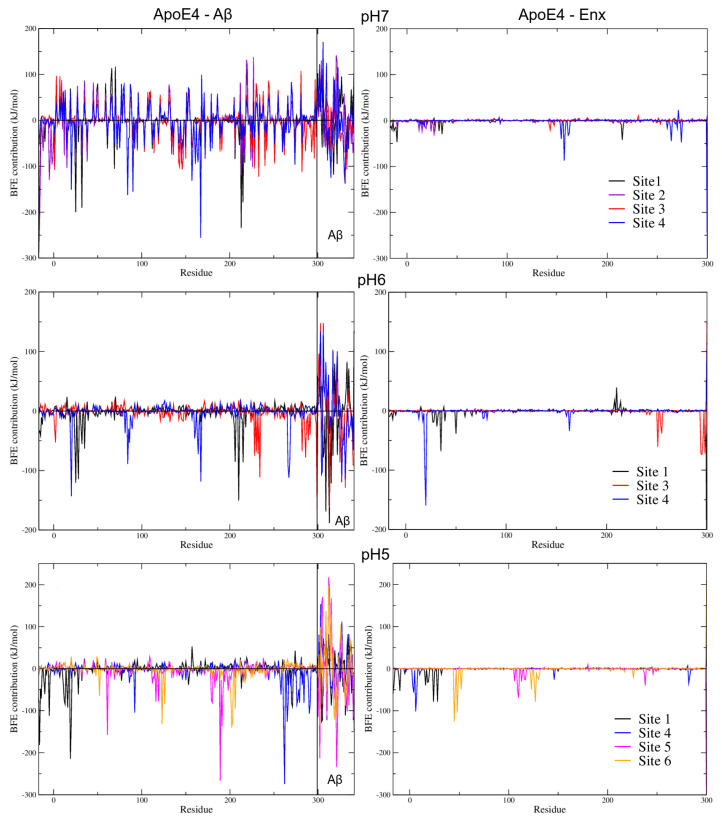
MM-PBSA calculation of BFE by residues for all ApoE4-Aβ-Enx complexes at the pHs studied. The left panels show the BFE per residue of the ApoE4-Aβ complexes with the highest energetic affinity for each interaction site. In the right panels the ApoE4-Enx complexes are shown. The color code used for each site was the same as that used for the docking sites, except for site S1, which is black for a better representation.

**Table 1 biomolecules-12-00499-t001:** Stability Descriptors of the ApoE4 and Aβ systems.

System	SASA ${}^{a}$	RMSD ${}^{b}$	RMSF ${}^{b}$	RG ${}^{b}$		H-Bonds	
				Total	Axis	Intra	Inter
**ApoE4**							
					x= 1.68 ± 0.15		
pH7	169.05 ± 4.02	0.61 ± 0.01	0.21 ± 0.12	2.08 ± 0.01	y= 1.72 ± 0.13	246 ± 10	746 ± 19
					z= 1.68 ± 0.16		
					x= 1.70 ± 0.17		
pH6	168.99 ± 6.59	0.59 ± 0.03	0.24 ± 0.12	2.09 ± 0.03	y= 1.69 ± 0.16	249 ± 11	728 ± 22
					z= 1.71 ± 0.14		
					x= 1.64 ± 0.13		
pH5	168.78 ± 5.20	0.67 ± 0.01	0.24 ± 0.15	2.03 ± 0.03	y= 1.67 ± 0.11	238 ± 9	732 ± 19
					z= 1.66 ± 0.11		
**Amyloid-β**							
					x= 0.81 ± 0.09		
pH7	33.17 ± 2.05	1.38 ± 0.01	0.35 ± 0.12	0.99 ± 0.08	y= 0.80 ± 0.09	20 ± 3	105 ± 6
					z= 0.81 ± 0.09		
					x= 0.88 ± 0.13		
pH5–6 rep-1	37.23 ± 2.27	1.16 ± 0.13	0.64 ± 0.16	1.10 ± 0.15	y= 0.91 ± 0.17	20 ± 3	105 ± 7
					z= 0.90 ± 0.15		
					x= 0.82 ± 0.09		
pH5–6 rep-2	34.21 ± 2.13	1.31 ± 0.14	0.53 ± 0.13	1.02 ± 0.10	y= 0.83 ± 0.13	18 ± 3	106 ± 7
					z= 0.83 ± 0.11		
					x= 0.87 ± 0.11		
pH5–6 rep-3	35.98 ± 2.08	1.29 ± 0.17	0.66 ± 0.18	1.06 ± 0.09	y= 0.87 ± 0.11	20 ± 4	106 ± 8
					z= 0.86 ± 0.10		

*^a^* In square-nanometers, *^b^* in nanometers. All values were obtained from the last 300 ns of the MD simulations.

**Table 2 biomolecules-12-00499-t002:** ApoE4 residues that are associated with the Aβ interactions.

	Site S1	Site S2	Site S3	Site S4
pH7	V-15, W-13, A-12, A-11, L-10, L28, R32, D35, E59, Q208, R213, L214, **R215**, A216	A-12, A-11, L-10, F-6, **L-5**, A-4, **G-3**, **C-2**, E9, P10, E11, P12, **E13**, **R15**, Q16, Q17, **T18**, E19, E27	**R142**, K143, K146, A241, E281, K282, V283, Q284, A285, V287, P295, **S296**, D297, N298, H299	W20, **T83**, **P84**, V85, **T89**, R92, L93, L93, D154, K157, R158, A160, V161, Y162, Q163, A164, R260, **S263**, **W264**, L268, Q270, **D271**, M272, R274
	**Site S1**		**Site S3**	**Site S4**
pH6	M-17, **K-16**, V-15, L-14, **W-13**, L-10, L-9, Q24, **R25**, **E27**, L28, A29, G31, **R32**, D35, R38, W39, E50, E59, E70, **R206**, **A207**, **Q208**, A209, W210, G211, **R213**, L214, M218		Q4, R142, D227, **R228**, **L229**, **D230**, E231, V232, K233, E234, R240, V287, G288, **T289**, N298, H299	E13, **W20**, **T83**, P84, K157, A160, V161, Y162, Q163, A164, G165, **R167**, **E266**, **P267**, L268, V269, E270, D271
	**Site S1**	**Site S4**	**Site S5**	**Site S6**
pH5	M-17, **K-16**, V-15, L-14, W-13, A-12, A-11, L-10, T-7, F-6, **L-5**, A-4, P12, R15, Q16, T18, **E19**, W20, **Q24**, R25, E27, L28	E109, R112, G113, V116, Q117, **R119**, G120, **R180**, L181, G182, P183, **R189**, **R191**, A192, A193, T194, Q204, L216, A237, E238, **K242**	E45, **Q46**, Q48, E49, L52, **Q123**, L126, G127, S129, P202, L203	**E7**, E13, Q258, L261, W264, F265, P267, L268, **V269**, **E270**, Q273, R274, W276, A277, G278, L279, V280, **K282**

Bold letters indicate electrostatic interactions.

**Table 3 biomolecules-12-00499-t003:** Average MM/PBSA free energies of ApoE4 complexes at pH = 7.0 in 100 ns of MD simulations.

Aβ Complex (site)	$\Delta E_{\mathrm{vW}}$	$\Delta E_{\mathrm{Elec}}$	$\Delta E_{\mathrm{PS}}$	$\Delta E_{\mathrm{SASA}}$	BFE
Sol1 (s4)	−247.04±2.38	63.41±13.31	261.49±8.71	−30.91±0.34	46.95±14.90
Sol2 (s1)	−239.07±4.15	−768.43±25.04	657.52±16.52	−38.02±0.41	−388.00±17.45
Sol3 (s4)	−349.41±2.26	276.96±9.88	261.34±6.21	−39.52±0.31	149.37±12.96
Sol4 (s4)	−392.38±2.60	−371.97±17.97	582.44±11.46	−47.70±0.36	−229.61±14.59
Sol5 (s4)	−275.51±4.99	−188.66±9.98	421.66±7.68	−36.43±0.46	−78.94±15.62
Sol6 (s2)	−252.40±3.14	175.35±18.95	308.17±9.62	29.50±0.41	201.66±17.79
Sol7 (s4)	−414.11±2.39	−237.80±11.07	563.52±7.20	−47.00±0.33	−135.39±12.47
Sol8 (s4)	−256.20±3.11	−233.35±9.05	526.61±6.47	−38.23±0.33	−1.17±11.88
Sol9 (s2)	−253.63±1.83	−252.00±12.40	391.33±7.55	−32.87±0.29	−147.16±13.77
Sol10 (s3)	−279.99±2.24	13.34±6.59	167.80±5.91	−38.60±0.32	−137.46±10.04
**Enx complex (site)**				
Sol1 (s4)	−297.07±1.86	−454.26±5.31	377.81±2.54	−31.49±0.10	−405.01±4.92
Sol2 (s2)	−58.43±7.68	−127.51±17.02	123.52±13.54	−7.28±0.97	−69.71±22.91
Sol3 (s1)	−125.70±6.48	−238.19±12.00	223.01±8.48	−19.65±0.95	−160.53±19.22
Sol4 (s3)	−120.17±15.14	−104.08±13.58	176.56±16.74	−14.13±1.75	−61.81±31.82

All values are in kJ · mol^−1^.

**Table 4 biomolecules-12-00499-t004:** Average MM/PBSA free energies of ApoE4 complexes at pH = 6.0 in 100 ns of MD simulations.

Aβ Complex (site)	$\Delta E_{\mathrm{vW}}$	$\Delta E_{\mathrm{Elec}}$	$\Delta E_{\mathrm{PS}}$	$\Delta E_{\mathrm{SASA}}$	BFE
Sol1 (s1)	−348.92±2.48	−1259.20±14.58	853.82±8.31	−50.68±0.31	−804.99±12.40
Sol2 (s1)	−390.00±2.76	−924.20±12.94	718.66±7.68	−55.61±0.33	−651.15±13.05
Sol3 (s4)	−395.02±1.79	−521.59±11.869	559.38±6.99	−48.58±0.21	−405.81±12.34
Sol4 (s1)	−234.84±2.80	−912.38±10.72	529.39±7.87	−36.01±0.32	−653.84±10.27
Sol5 (s1)	−287.39±3.22	−398.32±14.86	404.08±10.46	−39.56±0.25	−321.18±8.68
Sol6 (s4)	−212.45±2.07	74.62±12.33	78.11±6.09	−25.33±0.27	−85.05±9.49
Sol7 (s1)	−293.82±2.97	−1289.74±19.31	908.60±13.39	−48.39±0.36	−723.35±12.57
Sol8 (s1)	−302.66±2.27	−687.54±8.64	481.22±5.15	−44.02±0.27	−552.99±7.46
Sol9 (s1)	−214.97±2.35	−918.38±20.75	702.90±11.45	−42.88±0.32	−473.32±11.52
Sol10 (s3)	−235.48±2.21	−726.55±15.55	552.21±10.08	−32.44±0.31	−442.26±11.41
**Enx complex (site)**				
Sol1 (s1)	−199.68±2.04	−484.29±6.26	374.73±2.74	−29.33±0.14	−338.56±6.18
Sol2 (s3)	−124.20±1.76	−164.77±4.93	178.71±3.48	−15.60±0.15	−125.86±5.43
Sol3 (s4)	−89.41±2.41	−214.54±8.45	170.55±5.48	−13.08±0.28	−146.47±8.86

All values are in kJ · mol^−1^.

**Table 5 biomolecules-12-00499-t005:** Average MM/PBSA free energies of ApoE4 complexes at pH = 5.0 in 100 ns of MD simulations.

Aβ Complex (site)	$\Delta E_{\mathrm{vW}}$	$\Delta E_{\mathrm{Elec}}$	$\Delta E_{\mathrm{PS}}$	$\Delta E_{\mathrm{SASA}}$	BFE
Sol1 (s1)	−341.32±3.30	−839.35±17.73	729.14±10.62	−46.75±0.35	−498.28±16.12
Sol2 (s1)	−245.79±1.82	−626.532±15.03	398.32±9.04	−30.59±0.26	−504.59±12.19
Sol3 (s1)	−226.53±2.47	−248.28±14.81	190.27±11.32	−30.22±0.31	−314.76±11.51
Sol4 (s1)	−289.86±2.34	−1377.10±18.19	599.29±10.57	−38.22±0.24	−1105.89±13.05
Sol5 (s4)	−270.41±2.51	−734.13±14.88	559.90±9.93	−36.28±0.32	−480.91±14.01
Sol6 (s5)	−347.44±3.57	−810.80±18.25	720.30±11.16	−53.68±0.39	−491.62±14.45
Sol7 (s4)	−332.45±3.25	−1006.96±16.35	675.60±10.37	−44.86±0.34	−708.68±13.12
Sol8 (s5)	−347.27±3.23	−1562.07±16.01	1036.32±9.98	−58.03±0.34	−931.05±11.13
Sol9 (s6)	−168.80±1.99	−97.48±10.23	125.03±7.38	−22.63±0.31	−163.88±11.18
Sol10 (s1)	−235.89±2.73	−1143.97±9.66	697.82±12.18	−38.29±0.38	−720.23±13.09
**Enx complex (site)**				
Sol1 (s1)	−163.96±1.26	−295.52±3.46	282.29±2.46	−22.51±0.13	−199.69±4.40
Sol2 (s4)	−90.77±3.98	−314.55±10.13	243.74±7.03	−15.22±0.44	−176.80±10.67
Sol3 (s5)	−211.65±2.44	−611.97±7.23	431.08±4.95	−31.32±0.13	−423.86±4.72
Sol4 (s6)	−160.54±2.61	−313.53±9.53	280.77±7.80	−21.83±0.32	−215.12±9.31

All values are in kJ · mol^−1^.

**Table 6 biomolecules-12-00499-t006:** Top 10 residues that contribute to the binding free energy in the ApoE4 and Aβ protein structures at the S1 site.

No.	pH7			pH6			pH5		
	ApoE4${}_{A\beta }$	Aβ	ApoE4${}_{\mathrm{Enx}}$	ApoE4${}_{A\beta }$	Aβ	ApoE4${}_{\mathrm{Enx}}$	ApoE4${}_{A\beta }$	Aβ	ApoE4${}_{\mathrm{Enx}}$
1	M-17(−**292**)	F19(−**119**)	L-10(−**47**)	R25(−**126**)	H14(−**191**)	D35(−**67**)	E19(−**215**)	A42(−**147**)	Q24(−**78**)
2	R213(−**234**)	K16(−**119**)	R215(−**42**)	W210(−**121**)	Y10(−**160**)	E50(−**38**)	K-16(−**182**)	R5(−**126**)	L28(−**78**)
3	R25(−**200**)	D23(−**95**)	D35(−**29**)	L28(−**110**)	K16(−**132**)	G31(−**26**)	W20(−**128**)	H6(−**121**)	M-17(−**59**)
4	R32(−**191**)	V12(−**79**)	R32(−**23**)	R206(−**109**)	H13(−**106**)	E27(−**19**)	L-5(−**112**)	I32(−**111**)	L-10(−**53**)
5	R215(−**178**)	Q15(−**78**)	W-13(−**22**)	R32(−**85**)	D23(−**87**)	L28(−**19**)	Q16(−**92**)	K16(−**88**)	Q16(−**40**)
6	K-16(−**158**)	H6(−**59**)	V-15(−**20**)	R215(−**80**)	V18(−**76**)	W39(−**17**)	D271(−**91**)	L17(−**54**)	K-16(−**38**)
7	K69(−**105**)	L34(−**59**)	L28(−**18**)	E27(−**72**)	D7(−**68**)	W-13(−**14**)	E13(−**85**)	G33(−**48**)	T18(−**32**)
8	R217(−**93**)	V36(−**47**)	A-12(−**17**)	K69(−**52**)	S8(−**67**)	M-17(−**10**)	E270(−**82**)	A30(−**44**)	E19(−**31**)
9	R206(−**84**)	V18(−**41**)	M-17(−**16**)	D35(−**48**)	I41(−**57**)	E59(−**10**)	P12(−**76**)	G29(−**42**)	A-11(−**18**)
10	R226(−**83**)	E22(−**35**)	A-11(−**14**)	V-15(−**40**)	G25(−**41**)	R32(−**10**)	L-14(−**71**)	S26(−**31**)	R32(−**13**)

Values in bold and parentheses are BFE per residue in kJ·mol^−1^. Aβ and Enx suffix indicates the molecular complex.

## Data Availability

Not applicable.

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
