# Peer review of "In Silico Analysis of the Antagonist Effect of Enoxaparin on the ApoE4–Amyloid-Beta (Aβ) Complex at Different pH Conditions"

_biomolecules, 2022, doi:10.3390/biom12040499_

Round 1

Reviewer 1 Report

The manuscript by S. Paco-Coralla and colleagues is aimed at elucidating 7 molecular properties of ApoE4::Aβ interactions and the competition between Aβ and Heparin 8 for binding sites on ApoE4. The authors provided a first model of SLC22A6 and tried to characterize, via in silico approaches, its interactions with a set of chemicals. Moreover, Authors studied the pH effect on the stability of the ApoE4, Aβ and Heparin complexes by measuring the relative binding energy and the experimental binding free energies for the complexes.

Despite of the great interest of studies on these proteins, the manuscripts is affected by several major issues that should be carefully addressed before it can be considered for publication.

Major

  • Several statement must be improved, e.g.:
  1. Line 92-93: “The full-length ApoE4 protein was constructed taking the linear sequence obtained in the UniProtKB database, which has 317 residues, (Accession code: P02649.1).
  2. Line 145: “Periodic boundary conditions (PBC) in all directions.”
  • Line 206-208: “The ApoE4 protein was indicated as the “receptor” and the amyloid-β and heparin were indicated as the “ligand”.

Author must completely revise their manuscript, also improving English.

  • Authors must reorganize the section 2.1 of “Computational Details” to be clearer. In Line 84 Authors stated that “all the structures were modeled” but one of them is a crystallographic structure. Moreover, Authors must publish the sequence alignment between ApoE3 and ApoE4, highlighting the residues from ApoE4 and ApoE3 used for modeling. Since there are several Aβ structures available in the PRCS PDB, Authors must better justify why they chosen 1IYT. Move Figure 1 in the results.
  • Authors stated (Line 233) that all the systems converged after 300 ns, but in the line 234-235 they stated that the pH6 system is not yet fully converged. These two statements cannot “live together”. Authors must reorganize this section.
  • Since Aβis a peptide, to study the pH effect I strongly suggest performing some replicas of the “isolated Aβstructure”.

Minor

Authors must decide how to call the Amyloid-β and its abbreviation. In the manuscript it was called with different names.

Reviewer 2 Report

The article aims to analyze possible interactions between apolipoprotein E4 (ApoE4), β-amyloid (Aβ), and heparin using molecular docking and molecular dynamics simulations. Despite some methodological problems, the approach is interesting, and the results can potentially be valuable in understanding the pathological mechanisms of Alzheimer’s disease and designing of novel anti-AD drugs. The article can be published after a MAJOR revision addressing the following issues.

1) Most critically, the authors apparently used the heparin model structure with unionized sulfate groups (in Fig. 1, sulfate hydrogens are present and the molecular electrostatic potential is predominantly positive). This is obviously wrong, invalidating much of the further analysis. In addition, it should be noted that this structure is actually Enoxaparin, a synthetic low molecular weight heparin. The possible differences in its behavior compared to natural polymeric heparins should be discussed.

2) The results regarding the conformational behavior of Aβ (in particular, the disordered tangle structure) should be compared to other experimental and computational studies. It is also interesting to analyze what exactly is changing with pH, leading to such substantial changes in the structure. Only three His residues (with pKa about 6) in the N-terminal part are present in the sequence, thus their protonation states will undergo significant changes in the pH 5–7 range (from roughly 10% to 50% to 90% of the neutral form), and it is probably not possible to represent them with a single model for each pH. The phrase “surface with nucleophilic characteristics, denoted by the red coloration of its surface, which generates a greater interaction with negatively charged systems” is apparently confused.

3) In the final stage of analysis, the MD simulations of the docked ApoE4-Aβ complexes is performed only for 50 ns, and the full trajectory is used for the binding energy calculations. This duration is likely not sufficient for the full relaxation and refinement of the Aβ structure and binding mode, especially taking into account the flexibility of the disordered and tangled protein structure. (Among others, this could also explain the positive calculated interaction energies). In addition, the frequency of different docked poses is a very imprecise indicator of the “probability of complex formation”. The phrase “At pH7 conditions, positive energies were obtained in the ApoE4-Aβ interactions (non-interacting complexes)” is misleading as they are positive only for some of the complexes.

4) It is not clear what is implied by “measuring experimental binding free energies for the complexes” mentioned in the article.

5) In section 3.1, it is asserted that “The greater stability of the ApoE4 structure at pH6 can be explained by analyzing the hydrogen bonds formed” while both visual inspection of the plot and the summary values in Table 1 confirm that the differences are not statistically significant.

6) In Fig. 3, it would be helpful to overlay the structures at different pH in order to evaluate their differences. The faint “halo” around the structures in Fig. 3 and 4 (molecular surface?) is virtually invisible.

7) The Introduction, and especially the references therein, should be updated (many findings from 1990s and early 2000s likely have been refined and updated in the subsequent studies, especially in such a rapidly progressing field). The presentation of such results as “recent” also looks strange. The discussion of “the receptor-binding region in the N-terminal domain and by the lipid-binding region in the C-terminal domain” should be preceded by a general overview of the ApoE4 structure. It is also not clear if the Aβ accumulation is extracellular or intraneuronal. In Section 2.4, the part on molecular docking is misplaced. In Section 3.1.1, the explanations of the plot notation should be moved to the figure caption.

8) English in the article must be substantially improved with respect to misprints, terminology, grammar, and style. For instance, “considered to major the risk”, “initiation and progress of self-assembly by several disease-associated amyloidogenic proteins”, “Alzheimer’s disease (AD)… characterized by the loss of synapses”, “numerous biomarker candidates are being investigated, from structural, functional metabolic and neuroimaging, studies, including neuroimaging measures, sensory measures, digital biomarkers, blood levels of target proteins”, “a robust, accessible and potential biomarkers”, “protein-to-protein interaction”, “showed that under physiological conditions a small portion of Aβ interacts with ApoE lipoprotein particles in solution and the ability of ApoE lipoprotein particles to compete”, “lipidation”, “in the presence of low pH”, “based on shape complementary”, “which re-scoring”, “Rroot”, “all-length ApoE4 structure”, “regulator of neuronal plasticity and synapse”, “crystalline structure” (actually obtained by solution NMR), “this pH does not remain constant during the 500 ns of trajectory”, “With the best results”, “only an Aβ structure”, “NPT assembly”, etc. The notation is often inconsistent (Abeta, AB, AB42, alpha-helices, etc).

Reviewer 3 Report

The work presented by Paco-Coralla et al., is compelling and certainly of great interest for the community. Still, I think that the manuscript must be improved to accept its publication.

I have both, minor observations and major concerns. I'll start with the minor details:

I suggest revising the style and grammar along the manuscript, I provide some instances of flawed style or language:

Abstract Line 1: I believe that major is not the appropiate word here, please rephrase. Also, the term Aß is not defined, which may confuse potential readers not familiar with Alzheimer's. I suggest to improve the abstract and its presentation of the subject.

Line 15: "... of synapses leads" Incorrect grammar

Line 17: "AD is also associated with aging and it affects ..." Rephrase to improve the idea being introduced, the current wording is confusing as it seems disconnected.

Lines 20-27: I understand the main idea but the wording makes it difficult to follow. Consider rewriting this section.

Line 42: Change "The protein-protein interactions..." to Prtein-protein interactions between...

Also, there are several long sentences; e.g., lines 48-50: "However, a recent report showed that under physiological conditions a48
small portion of Aβ interacts with ApoE lipoprotein particles in solution and the ability of ApoE lipoprotein particles to compete with Aβ for cellular uptake via the low-density lipoprotein receptor-related protein 1 (LRP1) in astrocytes."

Lines 62-65 are confusing, please rephrase.

Line 68: "microenvirment"

Line 113: "generates"

Line 133: Change "with the OPLS" to ...using OPLS AA forcefield parameters

Section 3.4: NPT assembly

There are no details behind the rationale of some choices, for example the use of FireDock and OPLS forcefield. Also there is no mention of the version of OPLS used. Such details are important and must be stated.

Major comments:

There is no data on the quality of the homology model. Please provide this as supporting information.

One major concern I have is that the conclusions are based on one simulation. This is clearly not best practice there should be at least 3-5 replicas at each pH value. Such suggestion also applies to the BFE calculations, to better convey if convergence of energy is observed, if any.

Additionally, the authors state that convergence in some systems was not observed. With such observation and considering that RMSD was used as criterion, I suggest extending the simulations 200 ns to really achieve 500 ns of production runs.

In the discussion the authors mention the compaction degree of the protein. I think that it would be better if additional support such as RMSF plots are added to this discussion.

In Figure 3, the Circos plot is difficult to follow. Please consider making an individual figure for H-bond network and contact analysis.

Finally, the discussion of the BFE is plain. If possible include per residue contributions to the calculated energy. This can provide comparison points to previous statements and results (section 3.1).

I do think that the work is well presented, yet I strongly suggest to ensure that results are robust enough.

Round 2

Reviewer 1 Report

Authors addressed all the issues.

Reviewer 2 Report

The authors have significantly improved the article and addressed many issues identified by the reviewers. However, two critical issues have not yet been resolved, requiring a MAJOR revision of the article before it can be published.

1) The authors have clarified that the study uses Enoxaparin, a synthetic low molecular weight heparin; however, its role as a model structure, and the possible differences in its behavior compared to natural polymeric heparins, should be discussed more explicitly. But most critically, the study apparently still uses the structure with unionized sulfate and carboxylate groups (in Figs. 1 and 7, the sulfate and carboxylate protons are present, and the molecular electrostatic potential is predominantly positive, with small negative partial charges associated with hydrogen atoms). This is obviously wrong for pH 5–7, invalidating much of the further analysis.

2) Similarly, the Aβ protonation states should be analyzed more thoroughly. Unfortunately, the authors have not presented the PROPKA results in the Supplementary Materials, but a quick check with PROPKA 3.4 confirms the predicted pKa values of 6.36, 6.50 and 6.34 for His6, His13 and His14 residues, respectively. Thus, the pH 5–7 range is exactly where the most drastic changes happen in their protonation states (from roughly 97% protonated at pH 5, to about 70-80% at pH 6, to about 20-25% at pH 7). According to PROPKA, at the same time the total protein charge changes from +0.82 to –0.62 to –2.36. It is probably too difficult and unnecessary to consider all possible protonation states of the three residues, but at least a few representative structures at each pH should be modelled to evaluate their potential differences.

3) In several places, the Aβ structure (PDB: 1IYT) is referred to as “crystal” while it was in fact obtained by solution NMR spectroscopy.

4) In lines 287, 290, and 297, wrong units are listed (ns and nm instead of nm and nm2).

5) Minor stylistic and terminological mistakes should be corrected, such as “high electrostatic character”, “PES” instead of ESP for electrostatic potential, and “decreases a few”.

Reviewer 3 Report

The authors have made substantial improvements to their manuscript. Most of my concerns have been addressed.

I just have the following minor comments:

Several grammar errors persist, please revise the main text.

There are long sentences remaining in the text.

The presentation of pH values is inconsistent along the manuscript; i.e., pH5 or pH 5.

Check the typographical error on line 650. 

Figure SF2 has low resolution.

The formatting of tables makes difficult to follow ST8-13.

Check the grammar of the following table legends (ST14-16)
